# WHERE IS THE INVISIBLE: SPATIAL-TEMPORAL REASONING WITH OBJECT PERMANENCE

## ABSTRACT

Object permanence is a cognitive ability that enables humans to reason about the existence and location of objects that are not visible in the scene, such as those occluded or contained by other objects. This ability is crucial for visual object tracking, which aims to identify and localize the target object across video frames. However, most existing tracking methods rely on deep learning models that learn discriminative visual features from the visual context and fail to handle the cases where the object disappears from the image, e.g., occluded or contained by other objects. In this paper, we propose a novel framework for tracking invisible objects based on Qualitative-Quantitative Spatial-Temporal Reasoning (QQ-STR), inspired by the concept of object permanence. Our framework consists of three modules: a visual perception module, a qualitative spatial relation reasoner (SRR), and a quantitative relation-conditioned spatial-temporal relation analyst (SRA). The SRR module infers the qualitative relationship between each object and the target object based on the current and historical observations, while the SRA module predicts the quantitative location of the target object based on the inferred relationship and a diffusion model that captures the object's motion. We devise a self-supervised learning mechanism that does not require explicit relation annotations and leverages the predicted trajectories to locate the invisible object in videos. We evaluate our framework on a synthetic dataset (LA-CATER) and a new real-world RGB-D video dataset for invisible object tracking (iVOT) that contains challenging scenarios of human-object interactions with frequent occlusion and containment events. Our framework achieves comparable performance to state-of-the-art tracking methods that use additional relation annotations, demonstrating its generalization ability to novel scenes and viewpoints.

## 1 INTRODUCTION

Object permanence, a fundamental concept in developmental psychology (Baillargeon et al., 1985; Spelke, 1990), describes the ability of an agent to understand that an object persists even when the object becomes no longer observable. Studies (Aguiar and Baillargeon, 1999; Baillargeon and DeVos, 1991) have shown that early human infants only have this ability to understand occluded objects. As for accommodated objects, it takes a later age for infants to understand object permanence. As mentioned in Piaget's theory of cognitive development (Piaget, 2013), infants do not know that the world exists outside their concepts and experiences until they develop the concept of object permanence after 2 years old. This evidence demonstrates that understanding object permanence is a challenging task requiring thorough world modeling based on perceptual interactions with objects.

Establishing this concept in agents is essential for higher-level reasoning, as the objects observed by agents in complex natural scenes often occlude or contain each other dynamically. Therefore, object permanence is of great significance for guiding object tracking and even helping agents understand and model the objective world.

However, most reasoning methods Tokmakov et al. (2021); Zhang et al. (2021a); Ding et al. (2021) so far are only at the stage of understanding occluded objects, and a few methods Shamsian et al. (2020); Liang et al. (2021) that can track contained objects are still immature. For example, Tokmakov et al. (2021) use a spatial recurrent network (Ballas et al., 2015) to accumulate a representation of a scene and localize occluded instances by the representation while lacking an understanding of contained objects. OPNet (Shamsian et al., 2020) has the ability to localize fully invisible instances by using

two successive LSTMs (Graves and Graves, 2012) for predicting "who to track" and "where is it". However, the relations between the target and the container or occluder learned by OPNet are implicit and one-way, which is poorly interpretable and lacks an understanding of the global environment. AAPA (Liang et al., 2021) takes both detection results and action labels as input and constructs the accommodation hierarchy directly to guide target tracking. The method requires extra information and is limited in the application of real-world scenarios.

In order to establish accurate and interpretable spatial relations without using additional labels, we propose a novel Qualitative-Quantitative Spatial-Temporal Reasoning (QQ-STR) framework for tracking invisible objects inspired the object permanence. Our method proposes qualitative hypotheses of the spatial relations based on the historical trajectories of the objects and then verifies them according to the visual detection results. Thus, the quantitative results of the positions of targets can be derived by utilizing these hypotheses. First, we employ a vision module to localize visible objects and human poses in the image, which are regarded as visual abstractions. Then, a qualitative spatial relation reasoner (SRR) infers human-object and object-object relations based on visual abstraction, such as held by a hand, contained or occluded by an object. With that, a quantitative spatial relation analyst (SRA) based on the diffusion model estimates the location of invisible objects and outputs the bounding boxes in the image space. We demonstrate the effectiveness and robustness of our proposed method on both synthetic datasets and real-world datasets compared with state-of-the-art tracking models.

The main contributions of our work are as follows: 1) We construct a spatial relation representation for objects and simplify it into directed graphs with three types of relationships: "containment", "occlusion" and "no-direct-relation". The proposed paradigm is proved to be reasonable and significant for understanding object permanence. 2) We propose a Qualitative-Quantitative Spatial-Temporal Reasoning (QQ-STR) framework for tracking invisible objects and achieving state-of-the-art performances on both synthetic datasets and real-world datasets. 3) We collect a new real-world RGB-D video dataset iVOT for invisible object tracking that records daily indoor human-object interaction where objects often get occluded or contained. The dataset includes more complex spatial relations, such as nested containments and partial containments.

## 2 RELATED WORK

In this section, we review the most relevant works for learning object permanence in videos, which can be divided into two categories: *visual relational reasoning methods* and *visual datasets*. We first describe the existing methods for visual relational reasoning, which aim to infer the spatial relations between objects and track them across frames. Then, we introduce the existing datasets for evaluating visual relational reasoning, and compare them with our proposed dataset.

**Visual Relational Reasoning for Object Permanence.** Visual Relational reasoning aims to reason about the relationship between objects in the input image or video and to complete the specified object interaction or reasoning task. Existing methods can be further divided into rule-driven methods and data-driven methods. Rule-driven methods Liang et al. (2016; 2018); Wang et al. (2017); Liang et al. (2021) usually define the relationship between objects artificially and guide the visual reasoning task by some rules or assumptions. Liang et al. (2018) defines the accommodation relationship of objects and solves the target tracking task on the basis of Liang et al. (2016). Liang et al. (2021) takes both the detection results and motion labels as input and directly constructs the accommodation relationship hierarchy to guide the tracking task. Such rule-driven methods often require more information or assumptions to guide reasoning and have strong limitations. Data-driven approaches Tokmakov et al. (2021); Ding et al. (2021); Tokmakov et al. (2022); Wu et al. (2021) typically learn feature representations that can express relationships between objects from a large amount of training data, and use these features to guide visual reasoning tasks. For example, Ding et al. (2021) proposed a self-supervised method based on the transformer (Vaswani et al., 2017) structure to achieve visual question-answering tasks. Tokmakov et al. (2022) regards learning object permanence as fitting a random walk along memory and a target-focused self-supervised approach is proposed without exploring relationships between objects. The features learned by these data-driven methods are often implicit abstract, and poorly interpretable. The proposed QQ-STR method in our work combines the ideas of the two types of methods to have the best of both worlds. We define reasonable and explainable representations of spatial relations artificially and learn the spatial relations of objects in a self-supervised manner using only video annotations.

Table 1: Comparison of the video datasets used for visual relational reasoning.

| Dataset | Type | Containment | | Annotation | |
|---|---|---|---|---|---|
| | | Category | Hierarchy | Bounding Box | Spatial Relation |
| CATER (Girdhar and Ramanan, 2019) | Synthetic | 1 | ✓ | ✗ | ✓ |
| LA-CATER (Shamsian et al., 2020) | Synthetic | 1 | ✓ | ✓ | ✓ |
| PD (Tokmakov et al., 2021) | Synthetic | 0 | ✓ | ✓ | ✓ |
| Liang et al. (2018) | Real-world | 5 | ✗ | ✓ | ✗ |
| iVOT (Ours) | Real-world | 9 | ✓ | ✓ | ✓ |

**Visual Datasets for Learning Object Permanence.** As for evaluation, CATER (Girdhar and Ramanan, 2019), LA-CATER (Shamsian et al., 2020) and PD (Tokmakov et al., 2021) are widely used synthetic datasets for object interaction reasoning in videos. Since the PD (Tokmakov et al., 2021) dataset only includes simple occlusion relations, we chose the CATER and LA-CATER datasets for the evaluation. The synthetic datasets can be easily generated in batches using physics engines, but there are large gaps between synthetic scenarios and real-world scenarios. For example, there is an invisible hand manipulating the objects in the simulator, making the transition looks weird. Liang et al. (2018) dataset is a real-world RGB-D dataset with 44 short videos of 5 to 15 seconds. The major weakness of this dataset is that it contains merely simple spatial relationships, only occlusion and single-layer accommodation are included. In order to better evaluate the performance of our method in the real-world scene, we propose a new real-world RGB-D dataset iVOT(Invisible Visual Object Tracking) with more diverse scenes and more complex spatial relations, such as nested containment, mixed containments, etc. The iVOT dataset includes 9 container categories (cup, plate, fridge, hand, bag, drawer, sheet, bed, and sealed box), which is much more than existing datasets. Table. 1 shows the comparison of the proposed dataset with existing video datasets for visual relational reasoning.

## 3 METHOD

### 3.1 PROBLEM FORMULATION

We formulate the task as tracking a specific object in an input RGB video $\{I_1, I_2, I_3, ..., I_T\}, I_t \in \mathbb{R}^{3 \times H \times W}$ with a length of $T$. More specifically, we assume $T$ represents the length of the input video, $k$ represents the number of objects that appear in the video, an anchor for the $i$-th object at the frame $t$ is defined as $\{o_i^t = (id_i, bbox_i^t)\}_{i=1}^k$, where $id_i$ describes its identity, $bbox_i^t \in \mathbb{R}^4$ represents its bounding box. In order to describe the trajectories of targets over time conveniently, we use $\tau_i^*$ to represent the anchor sequence corresponding to the $i$-th object and use $\tau_i^*[l, r] = [o_i^l, o_i^{l+1}, \ldots, o_i^r]$ to represent the trajectories of the $i$-th object from the $l$-th frame to the $r$-th frame. Given the bounding box of the target in the first frame $\tau_{target}^*[1]$, the model is required to predict the bounding boxes of the target in other video frames $\tau_{target}^*[2, T]$, even if the object is invisible.

### 3.2 FRAMEWORK OVERVIEW

We view the object tracking task as studying how their spatial relations change over time, both when they are visible and after they become occluded or contained. As shown in Figure 1, our method decomposes the spatial-temporal relation reasoning into three steps, including the visual perception, the qualitative relation reasoning, and the quantitative trajectory prediction.

**The Visual Perception** is to extract abstract state, e.g., the location of the visible objects, from the raw-pixel images in the input video, which is the only input to the framework. Based on the visual abstraction we need judge whether the visible state of the target has changed and conduct the next stage of qualitative reasoning.

**The Qualitative Relation Reasoning** is used to reason the abstracted state and generate multiple hypotheses about the spatial relations, including human-object relations and object-object relations.

**The Quantitative Trajectory Prediction** deduces the events that lead to the change of the spatial relations and evaluates the possibilities according to the reasoning results of the last period. According to the historical trajectories of the objects, we infer the most likely spatial relations sets in the candidates and predict trajectories of invisible objects quantitatively.

As is shown in Figure 1, our method obtains the positions of visible objects and hands through the Visual Perception (represented as bounding boxes and stars at the bottom of the figure), and then

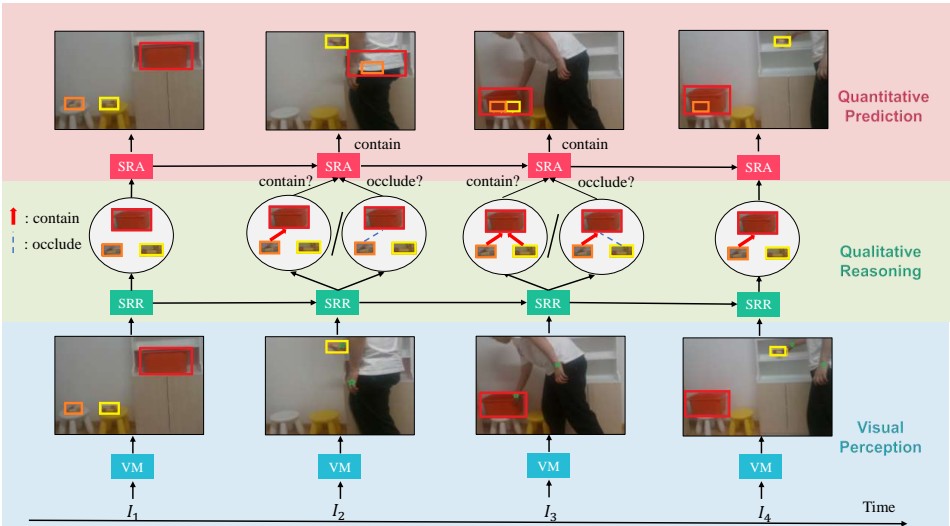

Figure 1: An overview of our proposed framework. We divide the spatial relation reasoning into three periods of analysis. Firstly, we extract the visual features of the input frames through the Visual Module (VM). Then the possible spatial relation graphs are assumed qualitatively by the Spatial Relation Reasoner (SRR). Finally, the trajectories of invisible objects are quantitatively inferred by the Spatial Relation Analyst (SRA). Please refer to Figure 4 ,Figure 5 and Figure 6 for detailed structures of these modules.

utilizes the Qualitative Relation Reasoning to generate corresponding spatial relation hypotheses (represented as circles in the middle of the figure). Finally, the method uses the Quantitative Trajectory Prediction to analyze the most appropriate hypothesis and generate the trajectories of the disappearing object (represented as new bounding boxes at the top of the figure).

Our model, shown in Figure 2, describes our method for tracking invisible objects from input raw video through causal inference of spatial relations. First, a vision module is employed to localize the visible objects and human poses for visual abstraction at every video frame, which is discussed in Section 3.3. Then, we describe our qualitative Spatial Relation Reasoner (SRR) that infers the human-object and object-object relations based on the visual abstraction in Section 3.4. With that, a quantitative Spatial Relation Analyst (SRA) estimates the possibilities for spatial relation candidates and predicts trajectories of invisible objects, which is detailed in Section 3.5.

## 3.3 VISUAL PERCEPTION

The visual module in our method consists of the detection module and the human-motion module, which aims to detect the visible objects and human poses in every video frame, regarded as a visual abstraction for further reasoning.

**Object Detection** is used to localize the visible objects based on the visual context. On each frame of the input video $\{I_1, I_2, I_3, ..., I_T\}, I_t \in \mathbb{R}^{3 \times H \times W}$, it outputs the identity $id_i \in \mathbb{I}$ and bounding box $bbox_i^t \in \mathbb{R}^4$ of all visible objects $o_i^t$ in the current $t$-th frame. For objects that are invisible in the current frame, we set $bbox_i^t$ to $\mathbf{0}$.

**Human Pose Estimation** is used when human-object interactions appear in the image. We use PoseC3D (Duan et al., 2022) to construct skeleton annotations for each person $j$ that appears. Then we only extract the key points of the human hands as a sequence of hand positions, donated as $H_t = \{(pl_1^t, pr_1^t), (pl_2^t, pr_2^t), \ldots, (pl_{h(t)}^t, pr_{h(t)}^t)\}$, where $h(t)$ represents the number of people detected in the $t$-th frame while $pl_j^t, pr_j^t \in \mathbf{R}^4, j \in \{1, 2, \ldots, h(t)\}$ represents the bounding box of the person $j$'s left and right hands in the $t$-th frame. Emperically, we simplify the hand as a special object that never becomes contained in our framework.

## 3.4 QUALITATIVE SPATIAL RELATION REASONER

In qualitative, the spatial relations between objects are defined as three types of relationships: **occlusion**, **containment**, and **no-direct-relation**. We regard each object as a node, and the spatial

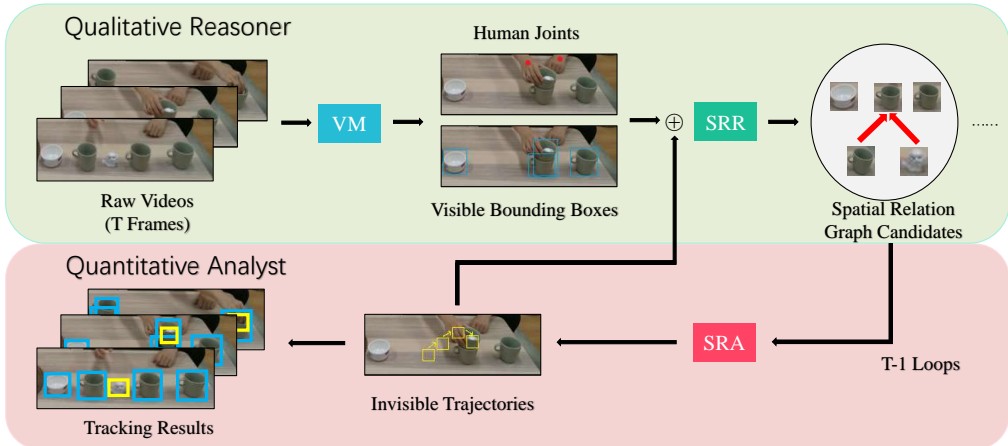

Figure 2: The architecture of QQ-STR. There are three key components: Visual Module (VM) which localizes the visible objects and human poses for visual abstraction, Qualitative Spatial Relation Reasoner (SRR) which infers the relationship between each object based on the historical observation and visible objects in the current frame, and Quantitative Spatial Relation Analyst (SRA) that estimates the location of invisible objects based on the inferred relations.

relations between each pair form a directed graph structure. The directed graph $G^t = \{occlude(i,j) \cup contain(i,j) \cup none(i,j), i \neq j, i, j \in \{1, 2, \ldots, k\}\}$ constructed in this way is defined the spatial relation graph of the current $t$-th frame.

In practice, there is very little information we can acquire about the contained and occluded objects, thus it is difficult to determine the only correct spatial relation graph corresponding to each video frame. Therefore, we generate at most $K$ spatial relation graphs $PG_t = \{G_1^t, G_2^t, \ldots, G_K^t\}$ in the current frame as candidates under each hypothesis. We define the trajectory of the object $j$ in the past $L$ moments under the assumption of the spatial relation graph $G_i^t$ at the $t - th$ moment as $\tau_j^{t,i} = [\overline{o_j^1}, \overline{o_j^2}, \ldots, \overline{o_j^k}]$. In addition, we will enumerate at most $K$ possible spatial relations in the first frame and form the candidates $PG_1$.

**The Spatial Relation Predictor** is used to predict what events cause changes in the visibility of objects. More specifically, let the identity of the object that changes state be $x$, the spatial relation predictor uses the historical trajectories $\tau = \{\tau_j^{t,i}[t - L + 1, t]\} \in \mathbf{R}^{L \times k \times 4}$ as input and predict the probability $f(\tau) \in \mathbf{R}^{k+1}$ that object $x$ is contained by a container or is occluded. The spatial relation predictor uses a Multi-head Attention model to learn the motion features of the objects relative to the target $x$ that may cause occlusion or containment events. The final probability vector is obtained through a multilayer perceptron.

**Training method.** We provide two approaches for training the spatial relation predictor. Supervised training will generate training trajectories $\{\tau_i \in \mathbf{R}^{L \times k \times 4}\}_{i=1}^M$ based on the motion labels of the training data and obtain the ground truth probability vectors $\{p_{\tau_i} \in \mathbf{R}^{k+1}\}_{i=1}^M$, that is, the containment or occlusion relationship corresponding to the motion label is 1, and the rest are 0.

As discussed in the previous question, the trajectories predicted by the proposed method will guide the search area of the tracker synchronously. In this way, the tracker is able to re-detect the object in the search area once the object reappears. During self-supervised training, our method uses only the original video as input, generates all possible spatial relation graphs when the target disappears. By utilizing the SRR module, the trajectories of the object under each assumptions are predicted. After that, when the object reappears later, we generate the corresponding probability vector by the intersection and union (IoU) of the bounding box of the reappearance position and the bounding box of the predicted position. The entire "disappear-reappear" segment is used as an episode of the training data for the SRA module.

**The Training Objective** for the qualitative Spatial Relation Reasoning module is defined as a simple cross-entropy loss:

$$\mathcal{L}_{reasoning} = \frac{1}{M} \sum_{i=1}^{M} -p_{\tau_i} \log(f(\tau_i)) \tag{1}$$

### 3.5 Quantitative Relation-conditioned Spatial-Temporal relation Analyst

After obtaining qualitative modeling of the spatial relations of objects, we can utilize the assumptions to infer crucial events and track invisible objects. The quantitative spatial relation analyst is used to analyze the possible spatial relation graphs $G_i^t$ of each frame and predict the trajectories of invisible objects under the hypotheses.

**Correction Module.** The correction module will check if the visual abstraction of the current frame is reliable. In the case of shivering or switching, it is assumed that the object coordinates have not changed. Let $bbox_i^t = (x_i^t, y_i^t, w_i^t, h_i^t)$ represents the bounding box of the $i$-th object at the frame $t$, we classify correctable object detection errors into two categories:

1. Shiver: $\exists i, (w_i^t, h_i^t) \neq (w_i^{t+1}, h_i^{t+1})$. The size of the bounding box of the object $i$ in the $(t+1)$-th frame does not match the size of the bounding box detected before. In that case of error, it is assumed that the object coordinates have not changed.

2. Switch: $\exists i \neq j, (x_i^t, y_i^t, w_i^t, h_i^t) \approx (x_j^{t+1}, y_j^{t+1}, w_j^{t+1}, h_j^{t+1})$ and $(x_j^t, y_j^t, w_j^t, h_j^t) \approx (x_i^{t+1}, y_i^{t+1}, w_i^{t+1}, h_i^{t+1})$. Two similar objects $i$ and $j$ are switched through the bounding boxes detected by the visual model. In that case of error, we swap the trajectories of the two detected objects.

**Trajectory Predictor.** Inspired by Tevet et al. (2022), we regard the trajectory prediction task as a condition-based generative task and try to apply the diffusion model. Under high uncertainty, the possible trajectories of the invisible target can be viewed as a noisy Gaussian distribution that represents a blurred area controlled by the container trajectory. As the uncertainty decreases, the distribution gradually approximates the ground truth distribution for generating the correct trajectory.

Diffusion is modeled as a Markov noising process. Given the trajectory $\tau_0$ drawn from the data distribution, the process aims to gradually add the indeterminacy to get the noise sequence $\{\tau_x\}_{x=1}^N$ until the ground truth trajectory is mixed with noise, where $N$ represents the number of the diffusion steps .

$$q(\tau_x|\tau_{x-1}) = \mathcal{N}(\tau_x; \sqrt{\alpha_x}\tau_{x-1}, (1-\alpha_x)\mathbf{I}) \tag{2}$$

$$q(\tau_x|\tau_0) = \mathcal{N}(\tau_x; \sqrt{\overline{\alpha_x}}\tau_0, (1-\overline{\alpha_x})\mathbf{I}) \tag{3}$$

Where $\{\alpha_x \in (0,1)\}_{x=1}^N$ are constant hyper-parameters and $\overline{\alpha_x} = \prod_{i=1}^x \alpha_i$. When $N$ is large enough and $\overline{\alpha_N}$ is small enough, we can approximate $\tau_N \sim \mathcal{N}(\mathbf{0}, \mathbf{I})$.

Next, we formulate the trajectory generation as a reverse diffusion process. For each possible spatial relation graph $\{G_i^t\}_{i=1}^K$ in the $t$-th frame, we enumerate all direct contained-container pairs $(c, container(c))$ from top to bottom according to the tree structure consists of containments. We formulate the reverse diffusion process for predicting each invisible object $c$ as follows.

$$p_\theta(\tau_{x-1}|\tau_x, f_{motion}) = \mathcal{N}(\tau_{x-1}; \mu_\theta(\tau_x, x, f_{motion}), \Sigma_\theta(\tau_x, x)) \tag{4a}$$

$$p_\theta(\tau_N) = G(f_{motion}) \tag{4b}$$

Where $f_{motion} = \tau_c^{t,i}[t-L+1,t] \cup \tau_{container(c)}^{t,i}[t-L+1,t] \in \mathbf{R}^{L \times 8}$ denotes the motion feature of $c$ and $container(c)$ in the past $L$ frames, $G(f_{motion})$ denotes a naive rule-based method to predict trajectories directly by historical motion features $f_{motion}$, $\theta$ denotes the parameter of the diffusion model. Instead of predicting $\epsilon_t$ as formulated by Ho et al. (2020), we follow Tevet et al. (2022) and directly predict trajectory itself $\theta(\tau_x, x, f_{motion})$. We use a perceptron as the diffusion model and train $\theta$ on ground-truth trajectories with a simple objective.

$$\mathcal{L}_{predict} = E_{\tau_0, x}||\tau_0 - \theta(\tau_x, x, f_{motion})|| \tag{5}$$

**Evaluation Module.** The evaluation module is used to calculate the probability of spatial relation graph $\{G_i^t\}$ in the $t$-th frame, so as to retain the most likely $K$ candidates and select the best one. Consider the process of obtaining the spatial relation graph $G_i^t$ by $t-1$ loops from the first frame, assume that the spatial relation predictor described in Section 3.4 have been performed $m$ times in

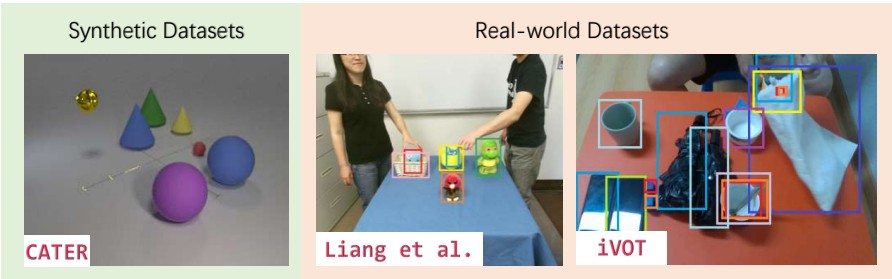

Figure 3: Video frames from CATER (Girdhar and Ramanan, 2019), LA-CATER (Shamsian et al., 2020), (Liang et al., 2018) and iVOT datasets.

the process, the predictions are $\{p_1, p_2, \ldots, p_m\}$ respectively, we define the probability of $G_i^t$ as $Pro(G_i^t) = \frac{1}{m}\sum_{j=1}^{m} p_j$.

In addition, we will correct $Pro(G_i^t)$ to 0 according to the results obtained by the visual module and reserve at most $K$ spatial relation graphs for each video frame according to the probability from high to low, as a set of candidate spatial relation graphs. The reappearance correction occurs when the IoU of the predicted bounding box and the position where the object reappears $< \lambda_{reappear}$, while the disappearance correction happens when the maximum IoU of the predicted bounding box and the bounding box of other visible objects $< \lambda_{disappear}$.

## 4 EXPERIMENTS

### 4.1 DATASETS AND EVALUATION

We conduct experiments on the following datasets and use the mean intersection over union (mIoU) metric for evaluation. Figure 3 show video examples of the following datasets.

**LA-CATER (Shamsian et al., 2020)** dataset is a synthetic video dataset generated on the basis of the CATER (Girdhar and Ramanan, 2019) dataset. More synthetic videos are generated according to the same generation rules, and the bounding box annotations are generated for each video, which is lacking in the CATER dataset. The LA-CATER dataset contains a total of $14k$ synthetic videos and is divided into 9300 training, 3327 validation, and 1371 test videos respectively.

**Liang et al. (2018)** is a real-world RGB-D dataset with 44 short videos about 5 to 15 seconds in 5 scenes containing 6k frames and 93 trajectories. The dataset contains simple occlusions and containments caused by human interactions.

**iVOT** is a real-world RGB-D dataset with $49$ long videos about $0.5$ to $1.5$ minutes in $12$ scene categories containing $31k$ frames and 171 trajectories, captured by Intel Realsense D435i. Each frame in the dataset is annotated with bounding boxes and actions by manual drawing combined with object tracking software. Our dataset contains more complex spatial relations, such as nested containments, mixed containments, occlusions, etc.

The comparison of the proposed dataset with existing video datasets for visual relational reasoning are shown in Table. 1. More details of these datasets are introduced in Appendix. A.5

### 4.2 IMPLEMENTATION DETAILS

We use the same network architecture as Shamsian et al. (2020) for object detection module in synthetic scenes, which uses a Faster RCNN (Ren et al., 2015) network pre-trained on the COCO (Lin et al., 2014) dataset as the backbone network, and the classifier head is fine-tuned on the LA-CATER (Shamsian et al., 2020) dataset. While for the experiments in real scenes, we apply AutoMatch (Zhang et al., 2021b), a recent object tracker, as the detector. During tracking, we limit the search area to the target's current location (detection position or predicted position). As for the human-motion module, we apply the PoseC3D (Duan et al., 2022) framework to extract human skeletons. The parameters used in the experiment are $L = 20$, $K = 16$, $N = 200$, $\lambda_{reappear} = 0.4$, $\lambda_{disappear} = 0.2$. The training for the Spatial Relation Reasoner (SRR) and the training for the

Table 2: Comparison of tracking methods on the test set of LA-CATER (Shamsian et al., 2020) with the same object detection and perfect perception setups using mean IoU. Since AAPA (Liang et al., 2021) uses additional motion annotations during testing, we only use it as a reference for constructing ideal relationships. The proposed QQ-STR reaches comparable performance to the state-of-the-art methods.

| Setup | Method | Visible ↑ | Occluded ↑ | Contained ↑ | Carried ↑ | Overall ↑ |
|---|---|---|---|---|---|---|
| OD | OPNet (Shamsian et al., 2020) | 88.98 | 80.19 | 77.07 | 56.04 | 82.35 |
| OD | PA (Liang et al., 2021) | 86.80 | 79.43 | 67.14 | 26.54 | 76.96 |
| OD | RAM (Tokmakov et al., 2022) | **91.70** | 79.30 | **82.20** | **63.30** | **83.14** |
| OD | Supervised QQ-STR (Ours) | 87.98 | **81.11** | 77.04 | 61.40 | 82.78 |
| OD | QQ-STR (Ours) | 87.87 | 80.85 | 75.42 | 57.12 | 81.76 |
| PP | OPNet (Shamsian et al., 2020) | 88.79 | 67.97 | 83.08 | 76.42 | 85.44 |
| PP | RAM (Tokmakov et al., 2022) | 91.43 | 82.65 | 86.37 | 75.18 | 89.43 |
| PP | Supervised QQ-STR (Ours) | **99.01** | **85.11** | **90.92** | **78.12** | **94.85** |
| PP | QQ-STR (Ours) | 98.61 | 84.03 | 89.65 | 76.41 | 93.76 |
| OD | AAPA* (Liang et al., 2021) | 88.67 | 82.15 | 80.79 | 68.25 | 84.66 |
| PP | AAPA* (Liang et al., 2021) | 99.38 | 90.53 | 93.86 | 82.54 | 96.31 |

Spatial Relation Analyst (SRA) are conducted separately. Both the training is performed with Adam optimizer with learning rates of 1e-4 (SRR) and 1e-3 (SRA) and the same batch size of 64. All the experiments are conducted on a GeForce RTX 3070 Lite Hash Rate. More implementation details are shown in Appendix. A.2.

### 4.3 Capturing Object Permanence in Ideal Synthetic Scenes

In this section, we evaluate the object permanence inferred by our algorithm on LA-CATER (Shamsian et al., 2020). Among the methods we compared, RAM (Tokmakov et al., 2022) and the proposed QQ-STR utilize no relationship annotations during training or testing. PA (Liang et al., 2021), OPNet (Shamsian et al., 2020), and Supervised QQ-STR use relationship annotations for training, but only raw videos for testing. As for AAPA (Liang et al., 2021), it uses relationship annotations for both training and testing thus we only use it as a reference for constructing ideal relationships.

The upper part of Table 2 shows the comparison of tracking methods with the same object detection setup. The proposed QQ-STR outperforms other methods in the "occluded" scenario and is slightly inferior to RAM (Tokmakov et al., 2022) in other scenarios. Our method achieves comparable performance to the state-of-the-art method, showing that the proposed framework can distinguish between containment and occlusion well when receiving non-perfect visual input.

The lower part of Table 2 shows the comparison of tracking methods with the same perfect perception setup, where the ground truth bounding boxes are used as the ideal detection results (note that the visibility of objects still depends on the result of object detection). Compared with the setup of object detection, the setup of perfect perception reduces the interference caused by the error of detection itself in reasoning the spatial relations and predicting the position of invisible objects, reflecting the model's performance ceiling under the assumption of using a perfect detection module. The proposed QQ-STR significantly outperforms OPNet (Shamsian et al., 2020) and RAM (Tokmakov et al., 2022) in every scenario but still has a gap with AAPA (Liang et al., 2021) that uses the ground truth spatial relations. A comparison of the two setups illustrates that our method is able to reason about spatial relationships more accurately under ideal detection results.

### 4.4 Learning Object Permanence in the Real World

In this section, we further demonstrate that our approach is able to discover object permanence in the real world, by conducting experiments on Liang et al. (2018) and the proposed iVOT dataset.

Table 3 shows our results on Liang et al. (2018) and the proposed iVOT dataset. Our method significantly outperforms Liang et al. (2018) and the detection baseline, but there is still room for improvement in the results of predicting the target state. That shows that learning object permanence in real-world scenes is more difficult than in synthetic scenes, including more complex spatial relations, human interactions, larger detection errors, and so on. Note that the results on the iVOT dataset are significantly worse than the results on Liang et al. (2018) because our proposed iVOT

Table 3: Comparison to the baselines on the validation sets of Liang et al. (2018) and the proposed iVOT with mIoU and state accuracy (i.e. determine if the object in the current frame is occluded or contained).

| Method | Liang et al. (2018) | | iVOT | |
|---|---|---|---|---|
| | mIoU ↑ | State Accuracy ↑ | mIoU ↑ | State Accuracy ↑ |
| Raw Detection | 0.570 | / | 0.491 | / |
| (Liang et al., 2018) | 0.674 | / | / | / |
| Supervised QQ-STR (Ours) | **0.748** | **0.719** | **0.583** | **0.686** |
| QQ-STR (Ours) | 0.711 | 0.699 | 0.542 | 0.614 |

Table 4: Analysis of the components of our method on the LA-CATER (Shamsian et al., 2020) dataset. The OD setup means using object detection while the PP setup means using the perfect perception. We ablate the number of the spatial relation graph candidates, whether to use action labels to construct ideal spatial relation graphs, and whether to use the diffusion-based trajectory prediction method.

| Setup | Candidates | SRR | SRA | Visible ↑ | Occluded ↑ | Contained ↑ | Carried ↑ | Overall ↑ |
|---|---|---|---|---|---|---|---|---|
| OD | $K = 1$ | w/o labels | with diffusion | 88.03 | 80.66 | 76.12 | 60.54 | 82.17 |
| OD | $K = 16$ | w/o labels | with diffusion | 87.98 | 81.11 | 77.04 | 61.40 | 82.78 |
| OD | $K = 16$ | with labels | with diffusion | **88.13** | **82.16** | **79.18** | **63.29** | **84.01** |
| OD | $K = 16$ | w/o labels | w/o diffusion | 88.03 | 80.77 | 77.09 | 61.16 | 82.62 |
| OD | $K = 64$ | w/o labels | with diffusion | 87.98 | 81.11 | 77.04 | 61.40 | 82.78 |
| PP | $K = 1$ | w/o labels | with diffusion | 98.47 | 81.67 | 89.35 | 77.03 | 93.58 |
| PP | $K = 16$ | w/o labels | with diffusion | 99.01 | 85.11 | 90.92 | 78.12 | 94.85 |
| PP | $K = 16$ | with labels | with diffusion | **99.42** | **90.12** | **95.21** | **86.10** | **97.04** |
| PP | $K = 16$ | w/o labels | w/o diffusion | 98.55 | 82.96 | 90.48 | 78.60 | 94.19 |
| PP | $K = 64$ | w/o labels | with diffusion | 99.01 | 85.11 | 90.92 | 78.12 | 94.85 |

dataset has more sophisticated spatial relations (i.e. nested containments, mixed containments, and occlusions), longer detection period, and more diverse scenes.

## 4.5 ABLATION STUDIES

We conduct diverse ablation studies on the LA-CATER (Shamsian et al., 2020) dataset to verify the effectiveness of the proposed method, as shown in Table 4.

**The Size of the Spatial Relation Graph Candidates.** First, we change the number $K$ of candidate spatial relation graphs generated by the spatial relation reasoner. The results in rows 1, 2, and 5 show that when $K$ reaches 16 and above, the candidates cover the optimal spatial relation graph, thus the performance is no longer improved.

**Verification for Spatial Relation Reasoner (SRR).** Next, we follow the method of AAPA (Liang et al., 2021) and use motion labels to construct ground truth spatial relation graphs, which are used to replace the candidates generated by spatial Relation reasoner. The results in rows 2 and 3 prove that the performance of the proposed method is close to the upper bound by using ground truth relations, demonstrating the effectiveness of the spatial relation reasoner.

**Verification for Spatial Relation Analyst (SRA).** In order to demonstrate the effectiveness of the trajectory prediction, we use a rule-based method based on the relative displacement to replace the diffusion model used in the Spatial Relation Analyst. The results in rows 2 and 4 demonstrate that the proposed diffusion-based trajectory predictor achieves better performance.

## 5 CONCLUSION

In this work, we propose a Qualitative-Quantitative Spatial-Temporal Reasoning (QQ-STR) framework for invisible object tracking, which defines the spatial relation between objects. Our framework achieves state-of-the-art performance on both synthetic and real-world datasets, demonstrating its ability to learn object permanence and infer spatial relations well.

For future work, we will improve the robustness of our method in some complex scenes, such as those with "occluded containers". Moreover, we will try to enable the model to learn the spatial relation representation between objects independently, without using artificial symbols.

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

## A    ADDITIONAL EXPERIMENTAL DETAILS

### A.1    SCHEMATIC DIAGRAMS

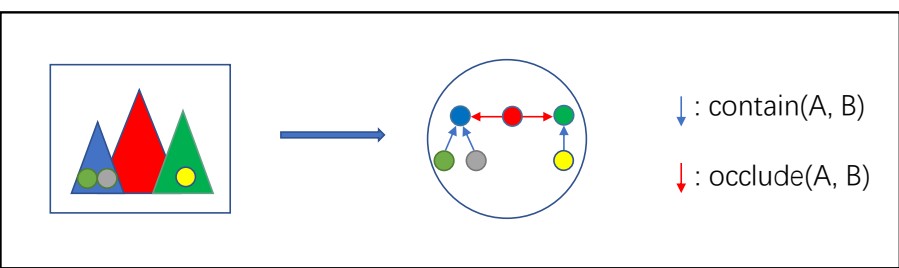

Figure 4: An example of a spatial relation graph. The scene on the left corresponds to the spatial relation graph generated on the right. For the sake of convenience, the non-direct relationships are omitted here.

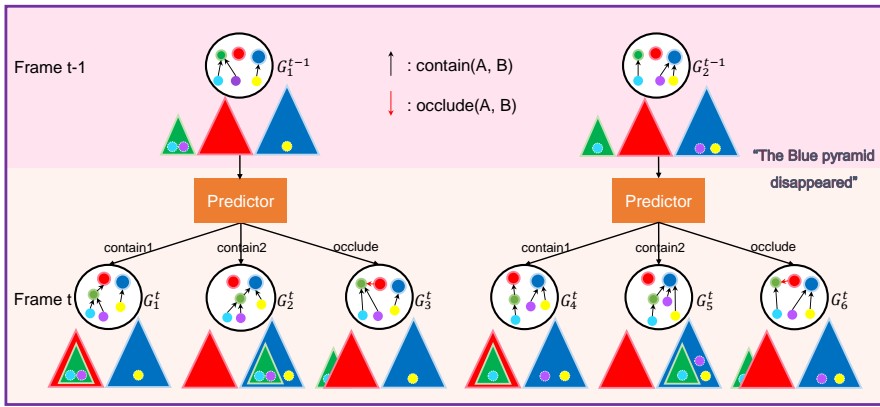

Figure 5: The structure of Qualitative Spatial Relation Reasoner. The spatial relation generator takes the spatial relation graphs of the previous frame as input and updates the corresponding spatial relation graphs in the next frame by the spatial relation predictor.

In order to represent the spatial relationship diagram constructed by our method more clearly, Figure 4 and Figure 5 respectively show a spatial relation graph example and the Qualitative Spatial Relation Reasoner schematic diagrams. As shown in Figure 5, the qualitative spatial-relation reasoner regards each spatial-relation graph $G_i^t$ from the previous spatial-relation graph candidates $PG_{t-1}$ as a node, and generates more possible branches $G_j^t$ in the current frame. These nodes constitute a search tree structure from top to bottom and are expanded in chronological order.

Table 5: Hyperparameters used in the proposed framework.

| Hyperparameters | Meaning | Value |
|---|---|---|
| $L$ | The length of the trajectories input into the predictor | 20 |
| $K$ | The maximum number of spatial relation graph candidates | 16 |
| $N$ | The steps of diffusion models | 200 |
| $\lambda_{reappear}$ | The threshold confidence of reappearance | 0.2 |
| $\lambda_{disappear}$ | The threshold confidence of disappearance | 0.4 |

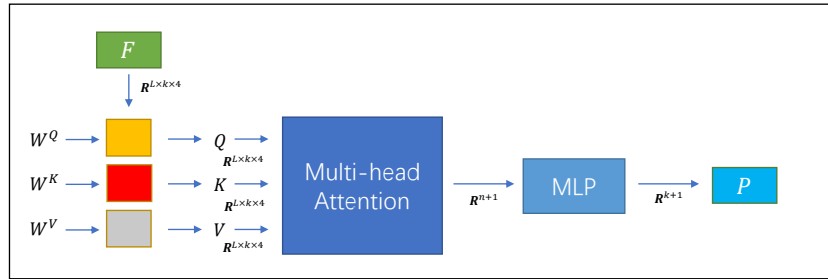

Figure 6: The network structure of the spatial relation predictor in Qualitative Spatial Relation Reasoner (SRR).

Table 6: The details of MLP for regressing probability vector.

| Layer | Type | Size |
|-------|------|------|
| 0 | Input | 15 |
| 1 | FC+ReLU | 256 |
| 2 | FC | 16 |

---

**Algorithm 1** Rule-based trajectory predictor

---

1: **Input:** Invisible object $x$ in the current frame $t$.
2: **Output:** Bounding Box Prediction $bbox_x^t = (x_x^t, y_x^t, w_x^t, h_x^t)$ of $x$ in the current frame $t$.
3: $y \leftarrow find\_container(x)$
4: **if** $x$ is visible in the last frame **then**
5:     $x1 \leftarrow (x_x^{t-1} - x_y^{t-1})/w_y^{t-1}$
6:     $x2 \leftarrow (x_x^{t-1} + w_x^{t-1} - x_y^{t-1})/w_y^{t-1}$
7:     $y1 \leftarrow (y_x^{t-1} - y_y^{t-1})/h_y^{t-1}$
8:     $y2 \leftarrow (y_x^{t-1} + h_x^{t-1} - y_y^{t-1})/h_y^{t-1}$
9: $x_x^t \leftarrow x1 \times w_y^t + x_y^t$
10: $y_x^t \leftarrow y1 \times h_y^t + y_y^t$
11: $w_x^t \leftarrow x2 \times w_y^t + x_y^t - x_x^t$
12: $h_x^t \leftarrow y2 \times h_y^t + y_y^t - y_x^t$
13: **Return** $bbox_x^t = (x_x^t, y_x^t, w_x^t, h_x^t)$

---

### A.2 IMPLEMENTATION DETAILS

**Spatial Relation Reasoner (SRR).** In this paper, we design a network consisting of a multi-head attention network (Vaswani et al., 2017) and a MLP for the spatial relation predictor in Qualitative Spatial Relation Reasoner, as shown in Figure 6. The length of the input trajectories $\tau$ is $L = 20$, and the maximum number of objects contained is $k = 15$. The $Q$, $K$, and $V$ used by the multi-head attention module are obtained from the input trajectories through three linear transformations, implemented by nn.Linear. The multi-head attention module is implemented by nn.MultiheadAttention with num_heads=5, embed_dim=15. The probability vector $p(\tau) \in \mathbf{R}^{n+1}$ is obtained by the last layer output of the multi-head attention module passing a MLP head shown in Table 6. The hyperparameters used in the proposed framework are listed in Table 5.

**Spatial Relation Analyst (SRA).** When predicting the trajectories of invisible objects in the Quantitative Spatial Relation Analyst, a simple rule-based method is used to make rough predictions. Then we use the motion feature of $x$ (trajectories predicted by the rule-based method) and $container(x)$ (visible trajectories or already predicted trajectories) in the past $l$ frames $f_{motion} \in \mathbf{R}^{l \times 8} (l = 5)$ and diffusion step $t$ as input and sampling the trajectory predictions $\tau^* \in \mathbf{R}^{l \times 4}$ after the denoising

Table 7: The details of MLP for the diffusion model.

| Layer | Type | Size |
|-------|------|------|
| 0 | Input | 41 |
| 1 | FC+ReLU | 64 |
| 2 | FC | 20 |

Table 8: Classification accuracy on the CATER (Girdhar and Ramanan, 2019) dataset using the metrics of Girdhar and Ramanan (2019). The proposed QQ-STR method achieves comparable performance with the SOTA methods.

| Method | Top1 Accuracy ↑ | Top5 Accuracy ↑ |
|--------|-----------------|-----------------|
| OPNet (Shamsian et al., 2020) | 74.8 | 90.2 |
| TPN-101 (Yang et al., 2020) | 65.3 | 83.0 |
| TSM-50 (Lin et al., 2018) | 64.0 | 85.7 |
| Aloe (Ding et al., 2021) | 74.0 | 94.0 |
| OCVT (Wu et al., 2021) | 76.0 | **94.4** |
| Loci (Traub et al., 2022) | **78.4** | 92.0 |
| QQ-STR (Ours) | 76.4 | 93.8 |

process through an MLP model as shown in Table 7. The parameters used in the diffusion process are $N = 200, \beta_1 = 1e - 3, \beta_T = 2e - 2, sampling\_times = 3$.

## A.3 TRAINING DETAILS AND EFFICIENCY

We train the Qualitative Spatial Relation Reasoner(SRR) and the Quantitative Spatial Relation Analyst (SRA) separately.

During training the spatial Relation predictor in the SRR module, we sample the "disappearance-reappearance" trajectories in the training set and obtain the corresponding probability vector as labels according to the self-supervised/unsupervised method. In this period, the model is trained with Adam optimizer for 200 epochs. The batch size and learning rate are set as $64$ and $1e - 4$ respectively.

While training the diffusion model for the SRA module, we use ground truth object trajectories in the training set as supervision for the denoising process. In each training epoch, we will randomly select a diffusion step $x \in [1, N]$ for each trajectory, and then obtain the mixed trajectory $\overline{\tau} = (\tau_0 + \epsilon_0, \tau_1 + \epsilon_0 + \cdots + \epsilon_1, \ldots, \tau_{x-1} + \sum_{i=0}^{x-1} \epsilon_i)$ by accumulating the temporal displacement Gaussian noise $\epsilon_i \sim \mathcal{N}(0, \mathcal{I} \times 0.1)$ to the initial position of the target $\tau_0$. The diffusion model is trained with a learning rate of $1e - 3$ for $500$ epochs.

In practice, our method achieves an average processing speed of 100 microseconds per frame when the size of the candidate set $K = 16$. As for the case where the size of candidate sets is larger or smaller, the processing speed of the proposed method is about 30 microseconds per frame ($K = 1$) and 250 microseconds per frame ($K = 64$).

## A.4 EXPERIMENTS ON THE CATER (GIRDHAR AND RAMANAN, 2019) DATASET

CATER (Girdhar and Ramanan, 2019) dataset is a synthetic video dataset for reasoning about object actions and interactions and contains a total of $5,500$ videos synthesized by the Blender engine. Each video consists of 300 frames and describes the movement of 5 to 10 objects on the table. Each object has 4 attributes shape (cube, sphere, pyramid, and inverted cone), size (small, medium, large), material (metal and rubber), and color (8 types in total). There is a small golden metal sphere appearing in every video, representing the target to be tracked. There are four types of behaviors in the moving process of an object on the table: sliding, rotating, lifting down, and containing. Since the CATER dataset does not have the annotation of the bounding boxes, we only conducted experiments on localization tasks on this dataset.

Table 9: The distribution of the iVOT dataset compared with the LA-CATER according to the spatial relation classification of the target.

| Dataset | Visible | Invisible | |
| --- | --- | --- | --- |
| | | Occluded | Contained |
| LA-CATER | 64.13% | 3.07% | 32.80% |
| iVOT | 40.98% | 42.04% | 16.97% |

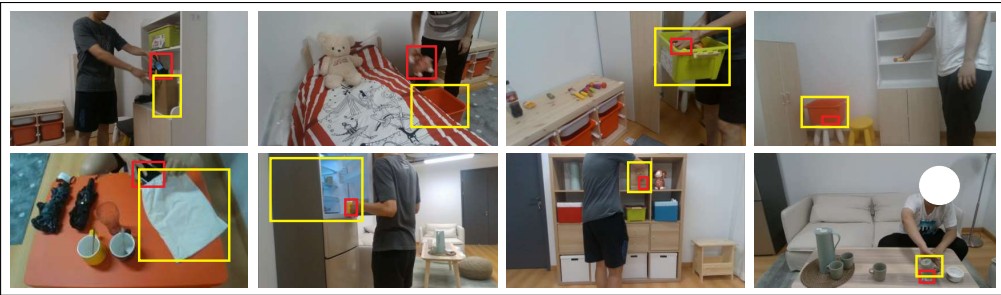

Figure 7: Several tracking examples in the proposed challenging iVOT dataset, where occlusions and containments occur frequently. The red bounding boxes represent the targets while the yellow bounding boxes represent the containers or obstacles.

Table 10: Comparison of the video datasets used for visual relational reasoning.

| Dataset | Action Patterns | Depth Information | FPS | Resolution | Scenes | Number of videos | Total frames | Video Length (seconds) |
| --- | --- | --- | --- | --- | --- | --- | --- | --- |
| CATER (Girdhar and Ramanan, 2019) | 3 | RGB | 30 | 320*240 | 1 | 5500 | 1650k | 10 |
| LA-CATER (Shamsian et al., 2020) | 3 | RGB | 30 | 320*240 | 1 | 4k | 4200k | 10 |
| PD (Tokmakov et al., 2021) | 1 | RGB | 20 | 1920*1080 | 630 | 630 | 6300 | 10 |
| Liang et al. (2018) | any | RGBD | 10 | 1920*1080 / 512*424 | 10 | 44 | 1346 | 5 - 15 |
| iVOT (Ours) | any | RGBD | 30 | 1920*1080 | 12 | 49 | 31k | 30 - 90 |

Table 8 shows the comparison results of our method with other localization methods on the CATER dataset. The proposed QQR-T achieves comparable performance with the existing state-of-the-art methods. Our method significantly outperforms the earlier methods like TPN-101 (Yang et al., 2020) and TSM-50 (Lin et al., 2018), slightly outperforms the latest methods like OPNet (Shamsian et al., 2020) or Aloe (Ding et al., 2021), and achieves comparable performance with OCVT (Wu et al., 2021) and Loci (Traub et al., 2022).

## A.5    DATASET

The proposed iVOT(invisible Object Tracking) dataset is collected by Intel Realsense D435i which has a RGB frame resolution of 1920*1080, a depth output resolution of 1280*720, and a frame rate of 30. The recording process is participated by two experimenters and the illumination conditions are all indoor white LED lights. Figure 7 shows several tracking examples in the challenging iVOT dataset, where traditional vision-based trackers have difficulty tracking targets (marked by the red bounding boxes) accurately. We show the distribution of the iVOT dataset in Table 9 according to the spatial relation classification of the target. According to the proportion of invisible frames, especially the proportion of occlusion frames, the dataset we proposed is far more difficult than LA-CATER. Table 10 is a comprehensive comparison of the proposed dataset with existing video datasets for visual relational reasoning.

## A.6    VISUALIZATION

We selected two typical tracking examples in the iVOT dataset for visualization. The results of AutoMatch (Zhang et al., 2021b), RTS (Paul et al., 2022), ToMP (Mayer et al., 2022), the proposed QQ-STR and the ground truth are displayed in chronological order from left to right. These two

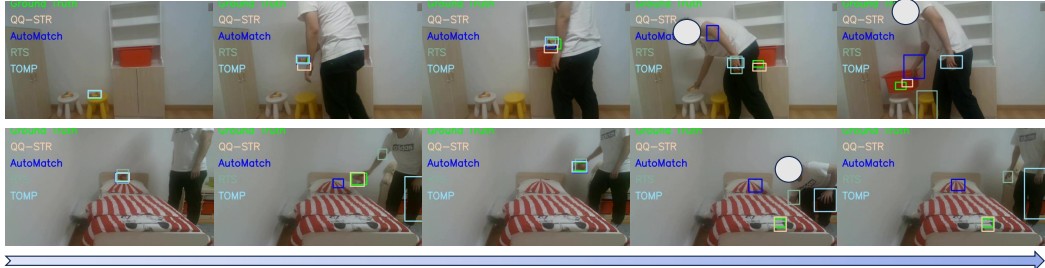

Figure 8: Two typical iVOT tracking examples with co-occurring containments and occlusions. The bounding boxes of five different colors represent the results of four different trackers and the ground truth. The proposed QQ-STR successfully predicts the trajectory of the disappearing target by constructing the spatial relation graph of invisible targets where current state-of-the-art trackers such as AutoMatch (Zhang et al., 2021b), RTS (Paul et al., 2022) and ToMP (Mayer et al., 2022) cannot.

Table 11: Comparison of the cross-domain generalization using mean mIoU. We train the models on the LA-CATER dataset and evaluate the performance on the iVOT dataset. We also report the results of OPNet as a reference.

|  | Trained on LA-CATER | Trained on iVOT |
|---|---|---|
| AutoMatch (Zhang et al., 2021b) | 0.491 | 0.491 |
| OPNet (Shamsian et al., 2020) | 0.476 | 0.535 |
| QQ-STR(Ours) | 0.554 | 0.583 |

tracking scenarios prove that the proposed QQ-STR understands the spatial relation of the invisible target well where traditional vision-based trackers cannot:

In the upper example, the experimenter removes the target toy from the stool and places the toy in a box on the bookshelf while keeping the toy occluded by the hands (Frame 1-3). Then he places another similar toy on another stool in the box (Frame 3-4). Finally, he moves the box from the bookshelf to the stool (Frame 4-5). In the first phase (Frame 1-3), all trackers track the visible target toy relatively efficiently. But when the box replaces the hand as the container of the target toy in the second stage (Frame 3-4), only QQ-STR successfully understands that the target toy has been contained in the box, while other trackers start to track the hands or even lose the target. In the third stage (Frame 4-5), our method successfully predicts the trajectory of invisible targets according to the trajectory of the box.

In the lower example, the experimenter removes the target doll from the bed (Frame 1-3), then places the doll under the bed (Frame 3-4), and finally stands up with empty hands (Frame 4-5). Other methods start to lose the target after the third frame while the proposed QQ-STR successfully understands that the doll is contained by the hand. Based on the fact that there is no target doll when the experimenter's hand reappears in the fifth frame, it is inferred that the bed becomes the new container for the doll in the fourth frame.

### A.7 SUPPLEMENTARY EXPERIMENTS

**Cross-domain Generalization.** We conducted experiments across datasets, that is, using the synthetic data set LA-CATER for training, and the iVOT data set only for testing. In this experiment, we use OPNet (Shamsian et al., 2020) for comparison and use the same AutoMatch (Zhang et al., 2021b) model as the vision module. Table 11 proves that our proposed method has stronger generalization ability and performance than OPNet.

**Hyperparameters Ablation Study.** In the ablation experiment, we left other hyperparameters unchanged and adjusted only one hyperparameter to test the impact of different hyperparameters on performance stability. Table 12 shows the robustness of the proposed method on different hyperparameter settings.

Table 12: Experimental results of our method on the LA-CATER dataset with different hyperparameters using mean IoU.

| Hyperparameters | Value | mIoU |
|---|---|---|
| $L$ | 10 | 82.10 |
| | 20 | 82.78 |
| | 40 | 81.78 |
| $K$ | 1 | 82.17 |
| | 16 | 82.78 |
| | 64 | 82.78 |
| $N$ | 100 | 82.63 |
| | 200 | 82.78 |
| | 1000 | 82.65 |
| $\lambda_{reappear}$ | 0.1 | 82.54 |
| | 0.2 | 82.78 |
| | 0.3 | 82.19 |
| $\lambda_{disappear}$ | 0.2 | 82.78 |
| | 0.4 | 82.78 |
| | 0.6 | 82.18 |

Table 13: Comparison of SOTA methods testing on the iVOT dataset using mean IOU.

| Method | mIoU |
|---|---|
| AutoMatch (Zhang et al., 2021b) | 0.491 |
| RTS (Paul et al., 2022) | 0.513 |
| ToMP-50 (Mayer et al., 2022) | 0.527 |
| ToMP-100 (Mayer et al., 2022) | 0.532 |
| QQ-STR(Ours) | 0.583 |

**SOTA Methods Testing on the iVOT.** Table 13 shows the performance of several SOTA trackers on the proposed iVOT dataset. Our method significantly outperforms current trackers without analysis of relationships between objects.

**Traditional Kalman Filter.** We applied the traditional Kalman filter method to conduct experiments both on the LA-CATER and iVOT datasets, and the experimental results in Table 14 showed that the traditional filter method cannot effectively understand the spatial temporal relationship under uncertain noise distribution and nonlinear system. That is the Kalman filter only predicts the short-term trajectories, and can not conditioned on the object-object relationship. However, the objects in containment may disappear in the long term.

Table 14: Comparison of the traditional Kalman filter method and our proposed method on the LA-CATER dataset and the iVOT dataset using mean IOU.

|  | LA-CATER | iVOT |
|---|---|---|
| Kalman filter | 0.6978 | 0.374 |
| QQ-STR(Ours) | 0.8278 | 0.583 |

