# OpenReview forum: "Where is the Invisible: Spatial-Temporal Reasoning with Object Permanence"
_ICLR.cc/2024/Conference — ICLR 2024 Conference Withdrawn Submission_

### Official Review · Reviewer_QThc · 2023-10-31

**Soundness:** 2 fair
**Presentation:** 3 good
**Contribution:** 2 fair
**Rating:** 5
**Confidence:** 5

**Summary:**

- This work proposes a novel method for tracking invisible/occluded objects in videos using object permanence from developmental psychology.
- The proposed method consists of three modules, namely, a Visual perception Module (VM), a qualitative spatial relation reasoner (SRR) and a quantitative relation-conditioned spatio-temporal relation analyst (SRA).
- The SRR module infers the spatial relationship between all the objects in a frame, i.e. whether the target object is occluded, using past and current information. The SRA module, modeled using a conditioned diffusion model, uses this information to predict the possible location of the target object in the future frame.
- Authors perform experiments on a synthetic and two real datasets and show competitive performance against contemporary methods.

**Strengths:**

- The use of diffusion models for tracking is interesting and useful to the community.

**Weaknesses:**

- Looking at a high level, the current method mimics the classical Kalman and Particle filter very closely (predict future location based on past observations and update the posterior based on current observations). Given the similarities, I believe comparing to such classical methods is necessary. I'm sure with the power of deep networks, the proposed method can outperform such classical methods but it is essential to know the gap in performance. Are the deep networks even necessary or does Kalman filter just solve the synthetic dataset?
- Authors demonstrate results on two real world benchmarks but I believe some more experiments are necessary to properly understand the contributions. RAM (Tokmakov et.al.) show results on KITTI benchmark. The ID-Switch problem in Multi-Object Tracking (MOT) is a result of models failing to understand object permanence. If authors claim their method is good at reasoning object permanence, it is essential to report results on these tracking datasets. I don't expect the authors to show state-of-the-art performance on Tracking/MOT but showing that this work improves some metric, like reducing the number of ID switches, is a good indicator of this method working on real world data. I strongly recommend authors to perform these experiments and report results on these tasks and datasets.
- Also consider reporting results on Occluded Video Instance Segmentation (OVIS) dataset [1] which deals with segmenting occluded objects in videos.
- A few design decisions haven't been ablated to understand their significance. Why did authors choose to go with diffusion models for object tracking? Why not go with the well tested trackers like SORT[2] or DeepSORT[3]? Or some recent state of the art MOT trackers [4]? I think this experiment is necessary to justify the use of diffusion models.
[1] Jiyang Qi, Yan Gao, Yao Hu, Xinggang Wang, Xiaoyu Liu, Xiang Bai, Serge Belongie, Alan Yuille, Philip H.S. Torr, Song Bai, Occluded Video Instance Segmentation: A Benchmark.
[2] Bewley, Alex and Ge, Zongyuan and Ott, Lionel and Ramos, Fabio and Upcroft, Ben, Simple online and realtime tracking.
[3] Wojke, Nicolai and Bewley, Alex and Paulus, Dietrich, Simple Online and Realtime Tracking with a Deep Association Metric.

**Questions:**

**Kindly address the concerns mentioned in the Weakness section for me to improve my rating**
- Not a major issue but make sure the tables and figures in the paper appear at the top of the page. This is probably my own preference but I would like the authors to consider this, to make the paper look more professional.

**Details Of Ethics Concerns:**

I do not foresee any immediate ethical concerns with this work.

---

> ### Author Response · Authors · 2023-11-22
> **Reply to Reviewer QThc**
>
> We wish to thank the reviewer for this helpful and comprehensive referee report, and we are glad to see your positive assessment of our methodology and presentation. Below we answer the questions and explain how the revised manuscript accommodates this detailed and helpful feedback.
>
> > *Q1. Looking at a high level, the current method mimics the classical Kalman and Particle filter very closely...*
>
> We applied the traditional Kalman filter method to conduct experiments both on the LA-CATER and iVOT datasets, and the experimental results showed that the traditional filter method cannot effectively understand the spatial temporal relationship under uncertain noise distribution and nonlinear system. That is the Kalman filter only predicts the short-term trajectories, and can not conditioned on the object-object relationship. However, the objects in containment may disappear in the long term.
>
> - Comparison of the traditional Kalman filter method and our proposed method on the LA-CATER dataset and the iVOT dataset using mean IOU.
>
> |                     | LA-CATER | iVOT |
> |---------------------|---------------------|-----------------|
> | Kalman filter               | 0.6978               | 0.374           |
> | QQ-STR(Ours)        | 0.8278               | 0.583           |
>
> > *Q2. Authors demonstrate results on two real-world benchmarks but I believe some more experiments are necessary to properly understand the contributions.*
>
> We agree that the evaluation of more real-world datasets is necessary. However, we notice that the object-object relationship in most existing real-world datasets is too simple to evaluate the object permanence reasoning. The invisible objects usually are just occluded by other objects or out of view, hardly contained by other objects. As a result, the invisible objects usually are shortly occluded by other objects, and the reasoning is unnecessary in most cases. However, in this paper, we focus on the ability to reason the localization of invisible objects, partially in some complex relationships, e.g., co-occurring containments and occlusions. Hence, this motivated us to collect new datasets in our daily activities that contain complex transitions of the object relationship. We argue that it will be our future work to collect more diverse videos with complex object relationships to evaluate the object permanence, rather than on those existing video tracking datasets, as the focus of the two domains are different.
>
> > *Q3. A few design decisions haven't been ablated to understand their significance. Why did the authors choose to go with diffusion models for object tracking? Why not go with the well tested trackers like SORT[2] or DeepSORT[3]? Or some recent state of the art MOT trackers [4]? I think this experiment is necessary to justify the use of diffusion models.*
>
> The main purpose of using the diffusion model is to predict the trajectories of invisible objects under certain spatial constraints rather than serving as a tracker itself. Under high uncertainty, the possible trajectories of the invisible target can be viewed as a noisy Gaussian distribution that represents a blurred area controlled by the container trajectory. As the uncertainty decreases, the distribution gradually approximates the ground truth distribution for generating the correct trajectory. Therefore, we regard the trajectory prediction task as a condition-based generative task and apply the diffusion model. The relationship experiments in Section 4.5 demonstrate the effectiveness of diffusion models for trajectory prediction.
>
> The MOT trackers also apply a similar tracking-by-detection framework. They focus on solving the data association problem for multi-object identification, where the objects may be of similar appearance and will shortly occluded by others. In this case, most MOT trackers assume the object keeps the linear dynamic (they usually track multiple pedestrians), and only apply some simple models to predict the short-term trajectory when the object is occluded by another. However, such short-term occlusion is a simple case in the study of object permanence.
> We find that the trajectory of the object highly depends on the relationship to other objects. If the object is only occluded by others, it will keep its own movements. In contrast, if it is contained in some object, its trajectory will depend on the transition of the container.
> That is why we need to build a relation-conditioned trajectory prediction for tracking invisible objects. The results of the Kalman Filter-based tracker also show that the simple predictor can not handle the complex relationship and long-term invisibility.
>
> > *Q4. Not a major issue but make sure the tables and figures in the paper appear at the top of the page.*
> >
> Thanks to the reviewer for valuable and helpful suggestions. We have revised the manuscript to make the Tables and Figures appear at the top of the page.

---

### Official Review · Reviewer_xz37 · 2023-11-01

**Soundness:** 3 good
**Presentation:** 2 fair
**Contribution:** 2 fair
**Rating:** 5
**Confidence:** 4

**Summary:**

This paper proposes a qualitative-quantitative reasoning framework for tracking invisible objects. The proposed method consists of three main modules, where a visual perception module is used to embed visual frames, SRR module is used to generate spatial relations between different objects, and SRA module predicts the location of the object based on the inferred relationships and a diffusion model. Experiments are performed on both synthetic and real-world datasets. Besides, this paper proposes a real-world RGB-D dataset, containing various scene categories, longer sequences, and more complex spatial relations.

**Strengths:**

1. This paper proposes a real-world RGB-D dataset, which might be beneficial for future research on the related task.
2. This paper proposes a novel qualitative-quantitative reasoning framework, separating the explicit qualitative reasoning analysis and the quantitative location prediction, which might provide a new sight to solve object permanence.

**Weaknesses:**

1. The different modules in the proposed framework don’t look compact, but independent. The framework integrates many separate models to address separate problems, like an object detector, a human pose estimation, the correction module, and a diffusion-model-based trajectory predictor. These different models are also trained separately, but not in an end-to-end manner. Compared with SOTA methods, results in Table2 are also not evident enough to show the advantages brought by this kind of complex design. Besides, the framework seems to bring many hyperparameters, I think it's necessary to explain these hyperparameter settings, and also perform corresponding experiments to show the robustness of the proposed framework on different hyperparameter settings.

2. Some detailed designs are not explained clearly. For example:
(1) For the object detection model, the classification of the general detection task is to identify different categories of objects, while I think the proposed method needs an object-level classification, the paper didn't mention much about this.
(2) For the shiver error in the correction module, what if the size of the object itself changes in adjacent frames, like aspect ratio change or scale change?

3. One of the main contributions of this paper is to propose a new dataset, however, there are very few descriptions of this new dataset. Besides, there is also a lack of experimental results of SOTA methods on the proposed dataset iVOT (Table3).

4. The paper is a bit hard to read since:
(1) the definition and use of some variables and formulas seem a bit complicated and have some typos. For example, the definition of hand positions H_i, the use of t/T/k, inconsistent use of superscript and subscript on different formulas, and so on.
(2) Fig1&2 are not connected tightly with text. The cases shown in the figures are not well explained in the text.

**Questions:**

Please find my concerns in the above "Weaknesses".
I am concerned most about issues 1,3 and 2 in order. I'll consider changing my rate if the authors explain them well.

---

> ### Author Response · Authors · 2023-11-22
> **Reply to Reviewer xz37**
>
> Thanks for your helpful and constructive suggestions. Below we answer the questions in detail.
>
> > *Q1: The different modules in the proposed framework don’t look compact, but independent. The framework integrates many separate models to address separate problems, like an object detector, a human pose estimation, the correction module, and a diffusion-model-based trajectory predictor. These different models are also trained separately, but not in an end-to-end manner. Compared with SOTA methods, the results in Table 2 are also not evident enough to show the advantages brought by this kind of complex design.*
>
> A1: We argue that the overall framework is well-organized and comprehensive, rather than independent. All the modules we introduced are essential for the reasoning of object permanence. The object detector localizes the visible objects, and the human pose estimation is to recognize the human-object interaction for identifying the change in the object-object relationship (as most object states are changed by human actions in the real world). To localize the invisible object, the trajectory prediction module proposes multiple possible trajectories conditioned on the reasoned spatial relationship. The correction module checks the reasonableness of the proposed trajectory based on the state of the visible objects. Each module is interdependent and indispensable for localizing the invisible objects, as shown in the ablation study.
>
> Compared to the end-to-end methods, our method has the following advantages:
> 1) **Generalizability.** The pre-trained model can provide generalizable state representation to make our framework easily transferable to unseen datasets. We have reported the cross-domain generalization of our method compared with the baselines, shown in the following Table. The models are trained on LA-CATER and tested on iVOT. Our approach demonstrates superior generalization capabilities compared to OPNet.
>
> </table>
> |                     | Trained on LA-CATER | Trained on iVOT |
> |---------------------|---------------------|-----------------|
> | AutoMatch            | 0.491               | 0.491           |
> | OPNet              | 0.476               | 0.535           |
> | QQ-STR(Ours)        | 0.554               | 0.583          |
>
> 2) **Modularity.** Each module (e.g., object detection, action recognition, trajectory prediction) can be easily replaced by state-of-the-art methods, thus our model can benefit from the advances of each component.
> 3) **Interpretability.** Our framework consists of modular components that can be individually analyzed and evaluated. For example, we can visualize the relationship graph and the forecasted trajectory produced by each module, which can help us identify the potential weaknesses and limitations of our approach.
>
> > *Q2: Besides, the framework seems to bring many hyperparameters, I think it's necessary to explain these hyperparameter settings, and also perform corresponding experiments to show the robustness of the proposed framework on different hyperparameter settings.*
>
> Thanks for your suggestions. We have provided the details about the hyperparameter in the Appendix. For convenience, we listed the main hyperparameters used in our method and conducted experiments to further analyze the effect of each hyperparameter in the LA-CATER. In the ablation experiment, we fixed other hyperparameters and varied only one hyperparameter to test the impact of different hyperparameters on performance stability. As is shown in the results, our framework can be robust to the change of hyperparameters. We have added this analysis in Section 4.3 and Table 2 of the revised manuscript.
>
> - Experimental results of our method on the LA-CATER dataset with different hyperparameters using mean IoU. We highlight the default value in bold.
>
> | Hyperparameters | Value | mIoU  |
> |-----------------|-------|-------|
> |                 | 10    | 82.10 |
> | $L$: The length of the trajectories input into the predictor            | **20**    | 82.78 |
> |                 | 40    | 81.78 |
> |                 |   |  |
> |                 | 1     | 82.17 |
> | $K$: The maximum number of spatial relation graph candidates              | **16**    | 82.78 |
> |                 | 64    | 82.78 |
> |                 |   |  |
> |                 | 100   | 82.63 |
> | $N$: The steps of diffusion models             | **200**   | 82.78 |
> |                 | 1000  | 82.65 |
> |                 |   |  |
> |                 | 0.1   | 82.54 |
> | $\lambda_{reappear}$: The threshold confidence of reappearance| **0.2**   | 82.78 |
> |                 | 0.3   | 82.19 |
> |                 |   |  |
> |                 | 0.2   | 82.78 |
> |$\lambda_{disappear}$: The threshold confidence of disappearance| **0.4**   | 82.78 |
> |                 | 0.6   | 82.18 |

---

> ### Author Response · Authors · 2023-11-22
> **Reply to Reviewer xz37**
>
> > *Q4: Some detailed designs are not explained clearly...*
> >
> A4: We are sorry that we did not describe the details exhaustively and caused your misunderstanding. We have revised the manuscript to explain more details. For convenience, we introduce the details in the following:
>
> *Object Detection*: For the experiments on the CATER and LA-CATER datasets, due to the limited types of objects included in the dataset, the vision module can be implemented by an object detector. For a fair comparison, we use the same object detector as previous works (OPNet[1], AAPA[2], and RAM[3]), a Faster RCNN network pre-trained on the COCO dataset, which is introduced in Section 4.2 of the paper. For the experiments in real scenes, we apply AutoMatch[4] as the object detector. During tracking, the search area in the next frame depends on the estimated target location (detection position or predicted position) in the current frame. This rule is widely used in visual tracking models. In this case, the adjustment of the search area will affect the performance of the tracker once it disappears.
> Hence, in our framework, once the confidence of the detection result is lower than a threshold of $\lambda_{disappear}$, we will assume the target has disappeared and predict the trajectory of the invisible target to guide the adjustment of the search area synchronously. In the same way, when the confidence of an object that originally disappeared is greater than $\lambda_{reappear}$, we will think that the object appears again. The proposed method focuses on predicting the spatial and motion relationships of objects in the invisible state, rather than vision-based object tracking.
>
> *Correction Module*:  Thanks for pointing this out. Since the frame rates of the LA-CATER synthetic dataset and the iVOT real-world dataset used are both 30fps, the scale of objects in adjacent frames changed slightly in most cases. If the scale changes dramatically, the correction module may result in unreasonable modifications. It will be our future work to extend our framework on videos with violent shaking.
>
> > Q5: *One of the main contributions of this paper is to propose a new dataset, however, there are very few descriptions of this new dataset. Besides, there is also a lack of experimental results of SOTA methods on the proposed dataset iVOT (Table 3).*
> >
> A5: We appreciate your suggestions. We have added more details and results about the new dataset. To be specific, the proposed iVOT(invisible Object Tracking) dataset is collected by Intel Realsense D435i which has an RGB frame resolution of 1920\*1080, a depth output resolution of 1280\*720, and a frame rate of 30. The data are recorded in indoor rooms (e.g., bedroom and living room), including diverse human-object interaction activities, where the objects frequently become invisible in the video. These cases are normal in our daily activities but are challenging to the state-of-the-art methods. For example, a man picks up a toy into a box and moves the box to another place, where the model needs to reason the object-object relation to localize the target object. More details on this dataset are elaborated in Appendix A.5. We also add more results of SOTA methods and visualizations in Appendix A.6.
>
> - Comparison of SOTA methods testing on the iVOT dataset using mean IOU.
>
> | Method       | mIoU  |
> |--------------|-------|
> | AutoMatch[4]    | 0.491 |
> | RTS[5]          | 0.513 |
> | ToMP-50[6]      | 0.527 |
> | ToMP-100[6]     | 0.532 |
> | QQ-STR(Ours) | 0.583 |
>
> > *Q6: The paper is a bit hard to read...*
> >
> A6: Thanks for your valuable and helpful suggestions. We have corrected relevant ambiguities or inconsistencies in the revision. We highlight the main update in blue.

---

> > ### Author Response · Authors · 2023-11-22
> > **Reply to Reviewer xz37**
> >
> > - References
> >
> > [1]. Shamsian, Aviv, et al. "Learning object permanence from video." Computer Vision–ECCV 2020: 16th European Conference, Glasgow, UK, August 23–28, 2020, Proceedings, Part XVI 16. Springer International Publishing, 2020.
> >
> > [2]. Liang, Ying Siu, Dongkyu Choi, and Kenneth Kwok. "Maintaining a Reliable World Model using Action-aware Perceptual Anchoring." 2021 IEEE International Conference on Robotics and Automation (ICRA). IEEE, 2021.
> >
> > [3]. Tokmakov, Pavel, et al. "Object permanence emerges in a random walk along memory." arXiv preprint arXiv:2204.01784 (2022).
> >
> > [4]. Zhang, Zhipeng, et al. "Learn to match: Automatic matching network design for visual tracking." Proceedings of the IEEE/CVF International Conference on Computer Vision. 2021.
> >
> > [5]. Paul, Matthieu, et al. "Robust visual tracking by segmentation." European Conference on Computer Vision. Cham: Springer Nature Switzerland, 2022.
> >
> > [6]. Mayer, Christoph, et al. "Transforming model prediction for tracking." Proceedings of the IEEE/CVF conference on computer vision and pattern recognition. 2022.

---

### Official Review · Reviewer_4XQC · 2023-11-02

**Soundness:** 3 good
**Presentation:** 2 fair
**Contribution:** 3 good
**Rating:** 5
**Confidence:** 4

**Summary:**

- The paper focuses on the problem of 2D bounding box-based object tracking under occlusion and containment.
The proposed approach is named QQ-STR, Qualitative-Quantitative Spatial-Temporal Reasoning. It has three components.
- a) Visual perception: Per frame object detection and human pose estimation using off-shelf methods.
- b) Qualitative spatial relation reasoning: Predicts the spatial relationship between objects in a frame and considers multiple possible object relationships as a graph by maintaining potential candidates.
- c) Quantitative relation-conditioned spatial-temporal relation analyst: Brings time into consideration. Error corrects and analyses the trajectory of the object. Helpful in tracking completely invisible objects.
- Evaluation is done on three datasets, LA-CATER, Liang et al. (2018), iVOT (collected by the authors).
- iVOT is RGB-D, 49 videos, 0.5 to 1.5 minute long - 12 scenes, 31k frames and 171 annotated trajectories.
- Baselines: OPNet (Shamsian et al. 2020), PA (Liang et al. 2021), RAM (Tokmakov et al. 2022), AAPA (Liang et al. 2021).
- The proposed method QQ-STR outperforms baselines on both synthetic and real datasets (mIoU metric).

**Strengths:**

- The paper is well-organized and easy to follow.
- The fundamental idea of using the graph structure for occlusion/containment reasoning for multi-object tracking is technically novel. The proposed modules make sense, and it is interesting to see the combination of the two components (qualitative and quantitative) working towards the state estimation of invisible objects.
- The experiments are done over multiple datasets, synthetic and real. The authors also contribute a dataset iVOT in this work, which is much appreciated. The scale of the dataset is also reasonably large compared to existing datasets focused on the same tasks.
- Ablations are done in Table. 4 to provide insights into the effectiveness of each components, SRR and SRA.

**Weaknesses:**

- No qualitative results: There is no single image in the main paper visualizing the tracking results of the proposed method.
Please consider showing multiple frames (which can be manually selected) of an object sequence and the tracked bounding boxes for QQ-STR and the closest baseline method (say RAM). This will tell us the improvements QQ-STR brings over the existing methods. If space permits, consider adding a failure case visualization highlighting the limitations. Note that the supplemental consists of two short videos (3 secs from Liang et al and 5 secs from LA-CATER) showing the results of the proposed method but no comparison to the baseline. It is a missed opportunity to not show video results on the collected iVOT dataset in the main paper or supplementary.

- Evaluation on the CATER dataset: Sec. 2 argues that the CATER dataset was not used in the evaluation as it only has classification and relation labels. However, in comparison to LA-CATER (the substitute), the CATER dataset is more widely used by current methods and has a more evolved list of quantitative performance of related methods. The results of the proposed method QQ-STR can still be evaluated on the CATER dataset, similar to the  CATER-Snitch localization task Table 1 of [1].

- Effect of number of objects on performance/Generalization: The proposed method considers all possible relations between the objects; the complexity of such a graph will increase when more objects come into the picture. This questions the ability of the method to generalize to conditions beyond five objects (which is the maximum used to show results). Testing on real-world tracking datasets like KITTIT (similar to RAM paper's Table. 2) would be a great way to showcase in-the-wild generalization of the proposed method.

- More descriptive method figures: Fig.1 and Fig.2 treat the proposed modules, VM, SRR, and SRA, as black boxes and tell nothing about the method. Please consider adding details about these components and provide insights into the inner workings of these blocks. Method figures are excellent visual tools to quickly convey the key ideas to the reader, which takes a while if it is only text-based (which is the case currently).

[1] LEARNING WHAT AND WHERE: DISENTANGLING LOCATION AND IDENTITY TRACKING WITHOUT SUPERVISION, ICLR 2023.

**Questions:**

As listed above,
1. Visual comparision of the QQ-STR and baseline.
2. CATER evaluation.
3. Going beyond the toy-object datasets, testing in-the-wild generalization.
4. Improved paper presentation (specific focus on the method figure).

---

> ### Comment · Reviewer_4XQC · 2023-11-22
>
> No reply from the authors. Keeping my original rating for "5: marginally below the acceptance threshold".

---

> ### Author Response · Authors · 2023-11-22
> **Reply to Reviewer 4XQC**
>
> We sincerely apologize for our delayed reply. We hope you will kindly reconsider your rating after reading our response. We appreciate your patience and understanding. We thank you for your helpful and comprehensive review. We have carefully addressed your questions and incorporated your feedback into the revised manuscript. The followings are the detailed response:
>
> > *Q1: No qualitative results...*
>
> A1: We agree with your comment that qualitative results are important to demonstrate the effectiveness of our method. We have added representative sequences on the iVOT datasets in the manuscript, as shown in Appendix A.6. These sequences compare the qualitative results of different methods and illustrate the advantages of our approach.
>
> > *Q2: Evaluation on the CATER dataset...*
>
> A2: We apologize for the confusion caused by our argument. We have evaluated our model on both the LA-CATER and the CATER datasets. Due to space limitations, we reported the results on the CATER dataset in Appendix A.3. We have clarified this point in the manuscript. In summary, our method outperforms most of the baselines in the Top 1 Accuracy and achieves comparable results to [1] (slightly lower in the Top 1 Accuracy but higher in the Top 5 Accuracy).
>
> > *Q3: Effect of number of objects on performance/generalization...*
>
> A3: We believe that this problem can be solved to a certain extent by increasing the size of the candidate relationship sets ($K$) or through appropriate pruning methods. In fact, in the LA-CATER and iVOT datasets of our experiments, the maximum number of objects is $15$ (more details are shown in Appendix A.2 of our paper), and on this scale of numbers, our proposed method shows satisfying performance. Finding a way to efficiently establish spatio-temporal relationships in crowded scenes is one of our future research directions.
>
> We also agree that the evaluation of more real-world datasets is necessary. However, we observe that most existing real-world datasets have simple object-object relationships that are not suitable for evaluating object permanence reasoning. As a result, the invisible objects are usually briefly occluded by other objects, and the relation reasoning is trivial in most cases. In contrast, in this paper, we focus on the ability to reason about the localization of invisible objects, especially in some complex relationships, such as co-occurring containments and occlusions. Therefore, we collected new datasets from our daily activities that contain complex transitions of object relationships. We argue that it is more meaningful to evaluate the object permanence on these new datasets, rather than on the existing video tracking datasets, as the two domains have different focuses.
>
> > *Q4: More descriptive method figures...*
> >
> A4: We appreciate your valuable suggestions. We have revised our manuscript to provide more details about each module of our method. Please refer to Appendix A.1 (Fig.4, Fig.5, Fig.6) for the detailed illustrations and explanations of our method.
>
> [1] LEARNING WHAT AND WHERE: DISENTANGLING LOCATION AND IDENTITY TRACKING WITHOUT SUPERVISION, ICLR 2023.

---

### Official Review · Reviewer_Z8r6 · 2023-11-09

**Soundness:** 2 fair
**Presentation:** 1 poor
**Contribution:** 2 fair
**Rating:** 5
**Confidence:** 4

**Summary:**

NOTE: My review is fully re-edited on Nov.10 because what was written before is about another paper.

The paper proposed a new method, QQ-STR, for invisible object tracking.
QQ-STR mainly contains of 3 modules:
- A Visual Module (VM) consisting of an object detector and a human pose estimator for visual perception from input video frames.
- A Spatial Relation Reasoner (SRR) that generates possible spatial relationship graph for objects in each frame at every timestamp
- A Spatial-temporal Relation Analyst (SRA) that predicts possible trajectories and select the best one as the prediction

Experiments are done on 3 datasets including one proposed by the authors themselves and the results suggest QQ-STR can achieve better or comparable performance as previous state-of-the-arts. Ablation studies support some important design choices.

**Strengths:**

- Experiments are done on 3 datasets and show seemingly competitive results
- Adapt a diffusion model for trajectory generation looks interesting
- Some ablation studies are provided to support a few important design choices.

**Weaknesses:**

- Many important details are not well explained or missing. For example, hand positions are extracted as in Sec. 3.3, but how it is being used is never mentioned in latter sections.

- Some parts don't make much sense. For example:
   - When generating spatial relationship graphs, do you distinguish occlusion and containment? If not, why?
   - Also, when generating the graphs, why do you only use object trajectories without identity information? The object characteristic should affect occlusion and containment, in my opinion.

- The method needs to "enumerate all possible spatial relations in the first frame and form the candidates PG1." This may cause some problems when there are too many objects.

- Despite a relative straightforward main idea of estimating occlusion / containment status based on past trajectories, there are a lot of twists and tweaks involved, for example, lots of hyperparameters and those correction stages, making the whole system over-complicated and vague to understand. The effects of most of them are unclear, and may hinder the generalizability to other datasets.

- The presentation needs to be improved. The whole idea is acutally Also, please double check inconsistent or incorrect notions, for example, "H_i = {(pl_i^t, pr_i^t), (pl_i^t, pr_i^t), . . . , (pl_i^t, pr_i^t)}" should be H_i = {(pl_i^1, pr_i^1), (pl_i^2, pr_i^2), . . . , (pl_i^t, pr_i^t)}

**Questions:**

Please address my concern according to the Weaknesses part.

---

> ### Author Response · Authors · 2023-11-22
> **Reply to Reviewer Z8r6**
>
> Thanks for the helpful and comprehensive review, and we are glad to see your positive assessment of our methodology ("Adapt a diffusion model for trajectory generation looks interesting") and experiments. Below we answer the questions and explain how the revised manuscript accommodates this detailed and helpful feedback.
>
> > *Q1: Many important details are not well explained or are missing.*
>
> We apologize for the lack of clarity and detail in our original manuscript. We have improved our manuscript to provide more details about our method. In particular, we have explained how the hand positions are used in the Spatial Relation Predictor (Section 3.4). Empirically, we regard the hand as a special object that never be contained by other objects in our framework. If we treat the hand as an ordinary object, it may cause ambiguity between containments and occlusions.
>
> > *Q2: When generating spatial relationship graphs, do you distinguish occlusion and containment? If not, why?*
>
>  When generating a spatial relation graph, we distinguish three types of spatial relationships: occlusion, direct containment, and no direct relationship. We use different edge categories to represent these relationships.
>
> > *Q3: when generating the graphs, why do you only use object trajectories without identity information? The object characteristic should affect occlusion and containment, in my opinion.*
>
> The visual module in our framework has already obtained the trajectories of visible objects by ID matching (using the same Faster RCNN network as OPNet and RAM on the LA-CATER dataset, and the AutoMatch tracker on real-world datasets). In this paper, the relation graphs focus on the relationship between invisible objects and visible objects. However, there is no identity information for invisible objects, so our method only uses historical trajectories without identity information when generating spatial relation graphs. The main goal of our work is to infer the trajectories of invisible objects by constructing the spatial relations of both visible and invisible objects, rather than using visual features to track visible objects.
>
> > *Q4: The method needs to "enumerate all possible spatial relations in the first frame and form the candidates PG1." This may cause some problems when there are too many objects.*
>
> We agree that the number of candidate relationship graphs may grow exponentially, so we only keep the most likely $K$ spatial relation graphs at each time step, as described in the Evaluation Module in Section 3.5. In practice, our method achieves an average processing speed of $100ms$ per frame when the size of the candidate set K=16. We have also tested the cases where the size of candidate sets is larger or smaller, and the processing speed of our method is about $30ms$ per frame (K=1) and $250ms$ per frame (K=64). We have updated the manuscript accordingly.

---

> ### Author Response · Authors · 2023-11-22
> **Reply to  Reviewer Z8r6.**
>
> > *Q5: Despite a relatively straightforward main idea of estimating occlusion /containment status based on past trajectories, there are a lot of twists and tweaks involved...*
>
> Compared to the end-to-end methods, our method has the following advantages:
> 1) Generalizability. The pre-trained model can provide generalizable state representation to make our framework easily transferable to unseen datasets. We have reported the cross-domain generalization of our method compared with the baselines. The models are trained on LA-CATER and tested on iVOT. Our approach demonstrates superior generalization capabilities compared to OPNet.
> 2) Modularity. Each module (e.g., object detection, action recognition, trajectory prediction) can be easily replaced by state-of-the-art methods, thus our model can benefit from the advances of each component.
> 3) Interpretability. Our framework consists of modular components that can be individually analyzed and evaluated. For example, we can visualize the relationship graph and the forecasted trajectory produced by each module, which can help us identify the potential weaknesses and limitations of our approach.
>
> To evaluate the generalization ability of our method to different datasets, we performed cross-dataset experiments, where we trained our model on the synthetic LA-CATER dataset and tested it on the real-world iVOT dataset. In this experiment, we use OPNet for comparison. The AutoMatch model is a part of the vision module for localizing the visible objects. Results show that our proposed method has stronger generalization ability and performance than OPNet.
>
> - Comparison of the cross-domain generalization using mean mIoU. We train the models on the LA-CATER dataset and evaluate the performance on the iVOT dataset. We also report the results of OPNet as a reference.
>
>
> |                     | Trained on LA-CATER | Trained on iVOT |
> |---------------------|---------------------|-----------------|
> | AutoMatch[1] | 0.491               | 0.491           |
> | OPNet[2]               | 0.476               | 0.535           |
> | QQ-STR(Ours)        | 0.554               | 0.583          |
>
> > *Q6: The presentation needs to be improved.*
>
> Thanks for your valuable and helpful suggestions. We have revised relevant ambiguities or inconsistencies in the paper.
>
> [1]. Zhang, Zhipeng, et al. "Learn to match: Automatic matching network design for visual tracking." Proceedings of the IEEE/CVF International Conference on Computer Vision. 2021.
>
> [2]. Shamsian, Aviv, et al. "Learning object permanence from video." Computer Vision–ECCV 2020: 16th European Conference, Glasgow, UK, August 23–28, 2020, Proceedings, Part XVI 16. Springer International Publishing, 2020.

---

### Author Response · Authors · 2023-11-22
**Revision Notes**

- Modified issues such as ambiguities, inconsistencies, and improper notations in the paper.
- Added more details like the explanation of architecture diagrams, the role of hand positions, and so on.
- Supplied diagrams of each module in Appendix A.1.
- Provided more details about the proposed iVOT dataset in Appendix A.5.
- Showed visualization comparison examples in Appendix A.6.
- Conducted more experiments in Appendix A.7.

---

### Meta-Review · Area_Chair_wzBv · 2023-12-04

**Metareview:**

This paper contributes an architecture for object tracking that is capable of handling objects that disappear from the image or are temporarily occluded. Taking inspiration from object permanence in humans, it is posited that this ability is critical for reasoning tasks from visual scenes. Experiments are conducted on synthetic and real-world datasets (including a new dataset: iVOT) where it is shown how the proposed architecture improves tracking performance over several baselines.

The reviewers highlighted several strengths of this work, such as the design of the model and the significance of the reported improvements, the technical novelty of using graph structures for capturing occlusion/containment reasoning for multi-object tracking, the dataset contribution, and the use of diffusion. On the other hand, there were substantial concerns regarding the clarity of the paper (many missing details), the complex design of the approach (introducing many hyper-parameters), and open questions regarding performance on real-world datasets. Some of these were (partially) addressed in the author's response, while others were not.

Ultimately this is a borderline paper, and it appears that a good number of reviewer issues stem from a lack of clarity regarding the initial version of the manuscript. The author's response was also only posted at the very last moment, which wasn’t helpful for having author-reviewer discussions either. In the reviewer discussion I have asked reviewers to take another look at the author's response and given them the opportunity to champion this paper. None of the reviewers opted to increase their score or champion this paper and as such I recommend a rejection.

**Justification For Why Not Higher Score:**

The reviewers were not sufficiently convinced by the author response to champion this paper or increase their score. Even so, this paper is borderline.

**Justification For Why Not Lower Score:**

N/A

---

### Decision · Program_Chairs · 2024-01-16

Reject